# Giant Atypical Neurofibroma of the Calf in Neurofibromatosis Type 1: Case Report and Literature Review

**DOI:** 10.3390/reports8030112

**Published:** 2025-07-17

**Authors:** Lyubomir Gaydarski, Georgi P. Georgiev, Svetoslav A. Slavchev

**Affiliations:** 1Department of Anatomy, Histology and Embryology, Medical University of Sofia, 1431 Sofia, Bulgaria; lgaidarsky@gmail.com; 2Department of Orthopedics and Traumatology, University Hospital Queen Giovanna-ISUL, Medical University of Sofia, 1431 Sofia, Bulgaria; 3University Hospital of Orthopedics “Prof. B. Boychev”, Medical University of Sofia, 1413 Sofia, Bulgaria; s.slavchev@medfac.mu-sofia.bg

**Keywords:** giant atypical neurofibroma, neurofibromatosis, diagnostics, treatment, clinical significance

## Abstract

**Background and Clinical Significance:** Neurofibromatosis type 1 (NF1) predisposes individuals to various peripheral nerve sheath tumors (PNSTs), including benign neurofibromas, malignant peripheral nerve sheath tumors (MPNSTs), and intermediate lesions known as atypical neurofibromatous neoplasms of uncertain biologic potential (ANNUBP), previously often termed atypical neurofibroma. These atypical lesions are considered premalignant precursors to MPNST. **Case Presentation:** We present the case of a 33-year-old male with NF1 who developed a rapidly growing, painful mass in his right calf. Clinical examination revealed signs consistent with NF1. Magnetic resonance imaging showed a large, heterogeneous mass in the lateral compartment. Biopsy revealed a neurofibroma with hypercellularity, moderate atypia, scarce S100 positivity, focal CD34 positivity, and an elevated Ki-67 proliferation index of 10–12%, consistent with ANNUBP. The patient underwent wide surgical resection, including the fibula and peroneal muscles. At the 30-month follow-up, there was no local recurrence, though the patient had a mild residual limp. **Discussion:** This case highlights the clinical presentation, diagnostic features, and management considerations for ANNUBP in NF1, emphasizing the importance of recognizing warning signs and the role of pathology in guiding treatment for these high-risk precursor lesions.

## 1. Introduction and Clinical Significance

Neurofibromatosis type 1 (NF1) is an autosomal dominant neurocutaneous disorder affecting approximately 1 in 2500–3000 live births [1,2]. It results from germline mutations in the NF1 tumor suppressor gene on chromosome 17q11.2. It leads to dysregulation of the Ras signaling pathway, predisposing patients to a broad spectrum of clinical manifestations—café-au-lait macules, intertriginous freckling, cutaneous neurofibromas, Lisch nodules, optic pathway gliomas, osseous dysplasias—and, notably, peripheral nerve sheath tumors (PNSTs) [2,3].

The PNST spectrum in NF1 spans benign neurofibromas (cutaneous, subcutaneous, plexiform) to highly aggressive malignant peripheral nerve sheath tumors (MPNSTs), which carry a lifetime risk of 8–16% and represent a leading cause of mortality in this population [3,4]. Plexiform neurofibromas occur in roughly half of affected individuals and are considered the primary precursors for MPNST development [5]. Historically, PNSTs in NF1 were dichotomized as benign versus malignant [6,7]; however, an intermediate category—atypical neurofibroma (AN) or atypical neurofibromatous neoplasm of uncertain biologic potential (ANNUBP)—has more recently been recognized. ANNUBP lesions exhibit atypical histological features (increased cellularity, nuclear atypia, mitotic activity) and often harbor CDKN2A/B loss, marking them as premalignant entities whose accurate identification and management are critical to preventing malignant transformation [8,9].

Herein, we present a rare case of a giant ANNUBP involving an entire muscle compartment in a patient with NF1. We detail the patient’s clinical presentation, imaging, histopathological findings, surgical management, and outcome, and provide a concise review of the existing literature on ANNUBP in NF1.

## 2. Case Presentation

A 33-year-old Caucasian male who had been formally diagnosed with NF1 presented to our institution with a fast-growing mass in his right calf with a duration of several months that had recently become painful and had started to cause difficulties in normal walking for the last several months. Clinical examination revealed multiple skin nodules and café-au-lait spots, consistent with NF1, on his face and trunk that were less pronounced in the extremities. The distal two-thirds of the lateral aspect of his lower leg was engaged by a firm, slightly tender, immovable mass with ill-defined margins. Additionally, the patient’s mother had died of a brain tumor at a young age.

Radiographs were non-specific. Magnetic resonance tomography revealed a heterogeneous mass confined within the lateral muscle compartment of the lower leg measuring 20 × 6 × 8 cm (Figure 1a,b).

After biopsy was performed, the tumor was found to have the phenotype of a neurofibroma with hypercellular areas with moderate atypia, with no irregular mitoses and no necrotic areas (Figure 2).

Immunohistochemical analysis revealed positive α-smooth muscle actin (α-SMA) staining primarily in vessel walls, highlighting the tumor’s vascularity. Diffuse and strong CD34 positivity is noted within the tumor cells, suggesting a vascular or mesenchymal origin, and its pattern helps assess architectural changes indicative of ANNUBP. The Ki-67 proliferation index is very low, 10–12% (calculated as the ratio of Ki-67 positive cells to all cells per 2 mm^2^), supporting a slow-growing nature. Nuclear p53 expression is largely negative, indicating no significant *TP53 gene* mutations, with its absence suggesting a less aggressive biological behavior compared to malignant progression. Focal S100 positivity is observed in scattered spindle cells, confirming their Schwannian lineage and suggesting potential neural or melanocytic differentiation. This collective profile, particularly the low Ki-67 and negative p53, aids in the diagnosis and classification of the mesenchymal neoplasm (Figure 3). Based on this, ANNUBP was diagnosed.

Surgery was performed through a lateral approach extending from the fibular neck to the lateral malleolus. Wide resection with negative margins was performed that encompassed the fibula and the peroneal muscles 8 cm distal to the tibiofibular joint, the anterior and the posterior intermuscular septum, the fibula as far as the tibiofibular syndesmosis, and the bellies of the peroneal muscles. The peroneal tendons were transected at the level of the syndesmosis, and their distal portions were fixed to bone behind the lateral malleolus in a plantigrade position of the foot. The wound was closed over a drain in the usual manner, and the lower leg was immobilized in a walking boot for six weeks. A control MRI found no local recurrence 30 months post-operation, as confirmed by Figure 1c. The patient walks without support with a mild limp.

## 3. Discussion

The present case illustrates the presentation and management of an ANNUBP in the extremity of an adult NF1 patient. The patient’s symptoms of rapid growth and new-onset pain are classic warning signs for potential malignant transformation or the development of an atypical lesion within a pre-existing neurofibroma in the context of NF1 [2]. Such symptoms warrant prompt investigation, often involving imaging and biopsy [10].

It is estimated that 8–16% of patients with NF1 develop malignant peripheral sheath tumors (MPNST) that mainly occur after the third decade of life [8]. MPNST is a well-established entity, and half of the cases occur in patients with NF1, which requires wide or radical resection [11]. However, ANNUBP have a less clear-cut profile. Based on genetic testing, some authors claim they are premalignant lesions in NF1 [12]. In contrast, others argue that they, along with low-grade MPNSTs, can be excised with positive surgical margins without compromising survival rates when preserving function is a priority [13]. In 2017, based on pathological and clinical data, Miettinen et al. [8] proposed a specific “clinical situation” termed ANNUBP, which was recognized by the World Health Organization in 2021 as a diagnostic entity synonymous with AN [14].

Several case reports and series describe ANNUBP in NF1 patients, often presenting similarly with pain or increasing size. Higham et al. reported on 76 ANFs in 63 NF1 patients (median age 27.1 years), where pain was the most common symptom (61%), and lesions were often palpable (59%) and appeared as distinct nodular lesions on MRI, frequently FDG-avid on PET scans [9]. Another case described a 32-year-old woman with NF1 and an ANNUBP of the sciatic nerve, also identified via imaging due to symptoms [15]. While many reported cases involve the trunk or central locations, extremity lesions, as in our case, are also well documented. The large size (20 cm) of the tumor in our patient is also a feature noted in other reports and considered a potential risk factor [10]. In this particular case of ANNUPB, it was considered that more “conservative” surgery would result in both loss of function of the peroneal muscles and positive surgical margins.

There is no generally accepted definition of a “giant” neurofibroma or ANNUBP. In the literature, there are examples of neurofibromas being labeled “giant” when measuring 5 × 5 cm in the neck and the foramen magnum, 8 × 8 cm in the proximal femur, or 145 × 40 cm and weighing 63 kg around the lower extremity [16,17,18]. Some authors have even tried to define the giant size of a tumor as a proportion of total body mass [19]. We find this approach problematic because it would mean that tumors of the same size would be “less giant” in patients with higher BMI than in patients with lower BMI. One might further speculate that in more obese patients, there would be virtually no giant tumors. Hence, surgery should be easier in obese patients since they have relatively minor tumors, which is absurd. Tumor size alone can pose a significant surgical challenge; however, it cannot be viewed separately from the tumor’s location and histological type. The authors believe that the term “giant” should be applied to lesions that either approach or exceed the size of their anatomical area, or when they greatly surpass the average size of a typically small tumor, e.g., osteoid osteoma or glomangioma.

The diagnostic process in this case relied on integrating clinical suspicion, imaging, and pathology. While non-specific, the MRI finding of a large, heterogeneous mass raised concern. Pathological examination was crucial. The diagnosis of ANNUBP was based on hypercellularity and moderate atypia. According to the 2017 consensus criteria, ANNUBP requires at least two of four features: cytologic atypia, hypercellularity, loss of neurofibroma architecture, and specific mitotic activity criteria (>1/50 HPF and <3/10 HPF) [8]. Our case met the criteria for atypia and hypercellularity. The immunohistochemical findings–scarce S100 and focal CD34—strongly suggest a loss of the typical neurofibroma architecture (specifically, loss of the CD34-positive fibroblastic network), likely fulfilling a third criterion [8]. The absence of irregular mitoses but a high Ki-67 index (10–12%) is noteworthy. While mitotic count is a primary criterion for grading ANNUBP versus low-grade MPNST (LG-MPNST requires 3–9 mitoses/10 HPF without necrosis), the elevated Ki-67 strongly indicates increased proliferation beyond a typical neurofibroma and aligns with findings in ANNUBP or even LG-MPNST. Ki-67 > 10% is often seen in MPNST but can overlap with ANNUBP. The weak, scattered p53 staining is less suggestive of high-grade MPNST, which often shows strong, diffuse positivity [8]. The findings firmly place the lesion within the ANNUBP category, representing a high-risk premalignant state.

The differential diagnosis of neurofibromas is paramount for the correct treatment. Clinically, neurofibromas present as soft, rubbery, often painless nodules tracking along peripheral nerves, which highlights the importance of correct differential diagnosis with schwannomas [20]. By comparison, schwannomas are encapsulated and more likely to cause focal neurological symptoms [20]. In contrast, MPNSTs grow rapidly, produce pain, and frequently arise in NF1 patients [21]. Similarly, lipomas feel doughy and demonstrate a uniform fat signal on MRI [22]. Moreover, epidermal inclusion cysts typically feature a central punctum and may discharge keratinous material [23]. On the other hand, DFSP presents as an indurated plaque rather than a discrete nerve-related mass [24]. Traumatic neuromas, conversely, are tender nodules that develop at sites of prior injury. Granular cell tumors arise in the dermis or submucosa and contain abundant eosinophilic granules [25]. Furthermore, desmoid tumors are firm, deep-seated masses often associated with β-catenin mutations [26]. Additionally, dermal neurotized nevi are pigmented lesions that stain for Melan-A [27]. Finally, myxoid tumors—such as intramuscular myxoma and aggressive angiomyxoma—feature abundant myxoid stroma, with the latter also expressing hormonal receptors [28]. On MRI, neurofibromas often display a central “target” sign and split fat sign with nerve entering/exiting centrally [29], while schwannomas show eccentric nerve entry and ancient variants may enhance heterogeneously [20,30], and MPNSTs demonstrate ill-defined margins with peripheral enhancement and low apparent diffusion coefficient (ADC) values [31]. Histologically, neurofibromas comprise a heterogeneous mix of Schwann cells, fibroblasts, and mast cells in a myxoid stroma with wavy nuclei and “shredded carrot” collagen [32], contrasting with the Antoni A/B areas and Verocay bodies of schwannomas [20], hypercellular herringbone patterns with pleomorphism in MPNSTs [21], storiform spindle cells in DFSP, and granular cytoplasm in granular cell tumors [24]. Immunohistochemically, neurofibromas show patchy S100 positivity, CD34 positive fibroblasts, and retention of intratumoral axons by neurofilament staining [32], whereas schwannomas are diffusely S100 positive and calretinin positive [33], MPNSTs often lose H3K27me3 and have variable S100 [21], DFSP is strongly CD34 positive [24], and dermal nevi express Melan A and S100 [27]. An in-depth comparison of the differential diagnosis of neurofibromas is presented in Table 1.

The therapeutic approach in the present case was wide surgical resection, including bone and muscle, aiming for maximal oncologic control. Throughout the literature, excision of ANNUBP is advised, as it may diminish the likelihood of malignant change [9]. Bernthal et al. demonstrated that even when margins remained positive following ANNUBP excision, recurrence rates were low, reinforcing the rationale for removing ANF pre-malignancy [13]. In Leuven, practitioners excise lesions without biopsy when there is no suspicion of high-grade MPNST, a strategy endorsed by the recent MPNST State of the Science consensus conference [21]. Establishing uniform histopathological criteria for ANNUBP assessment could facilitate the detection of lesions at elevated risk for malignant progression. Although the excision of numerous or deeply situated lesions poses challenges, surgical intervention remains advisable. For unresected ANNUBP, lifelong surveillance is imperative, and all patients should be regularly evaluated for the emergence of new ANNUBP [9]. In our case, the successful outcome with no recurrence at 30 months supports the effectiveness of the chosen surgical strategy in achieving local control for this specific high-risk lesion, albeit with a functional cost.

## 4. Conclusions

The present case demonstrates the clinical presentation, diagnostic evaluation, and surgical management of a giant ANNUBP in the lower extremity of a patient with NF1. The presence of rapid growth and pain served as crucial warning signs. Pathological assessment, including histology and immunohistochemistry (particularly Ki-67), confirmed a high-risk premalignant lesion consistent with ANNUBP. Wide surgical resection achieved local control, highlighting the effectiveness of surgery in managing these precursor lesions. However, the optimal extent of resection for ANNUBP remains debated, balancing oncologic safety and functional preservation. This case underscores the importance of vigilance, accurate pathological classification, and individualized surgical planning in managing complex PNSTs in NF1 patients to prevent progression to MPNST. Long-term surveillance remains essential due to the underlying NF1 predisposition.

## Figures and Tables

**Figure 1 reports-08-00112-f001:**
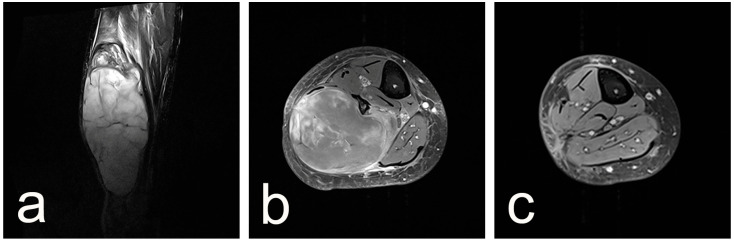
MRI of the right calf in a patient with a large soft tissue tumor. (**a**) Preoperative T2-weighted image in the frontal plane shows a well-circumscribed, multilobulated hyperintense mass occupying the posterior compartment of the calf. (**b**) Preoperative transverse T2-weighted image further delineates the extent and internal heterogeneity of the lesion. (**c**) Postoperative transverse image demonstrates complete resection of the mass with preservation of adjacent muscular and neurovascular structures.

**Figure 2 reports-08-00112-f002:**
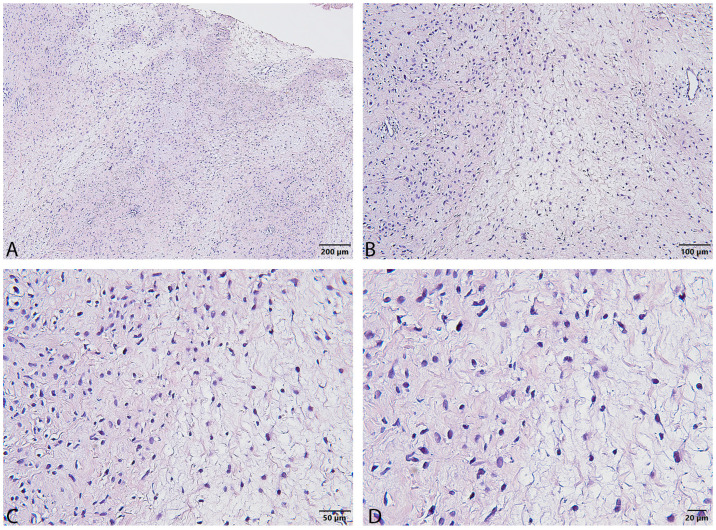
Histopathological images of a neurofibroma displaying hypercellular areas with moderate nuclear atypia, without evidence of irregular mitotic figures or necrosis. Hematoxylin and eosin staining (**A**–**D**); Scale bar 200 µ (**A**); 100 µ (**B**); 50 µ (**C**); 20 µ (**D**).

**Figure 3 reports-08-00112-f003:**
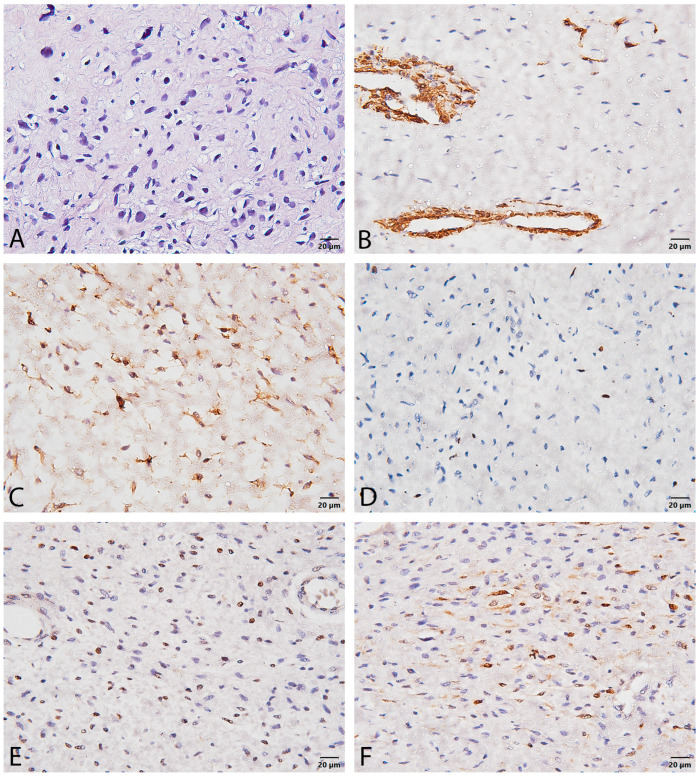
Higher magnifications of hematoxylin and eosin staining (**A**) and immunohistochemical staining of tumor tissue demonstrating the expression of various markers: (**B**) α-smooth muscle actin (α-SMA); (**C**) CD34; (**D**) Ki-67; (**E**) p53; (**F**) S100. Scale bar 20 µ.

**Table 1 reports-08-00112-t001:** Comparative features of neurofibroma and its differential diagnosis, summarizing their clinical presentation, imaging characteristics, histological appearance, and immunohistochemical profiles.

Tumor	Clinical Features	Histological Appearance	Immunohistochemical Appearance
**Neurofibroma**	Rubbery, often painless cutaneous or subcutaneous nodules; may cause itching, tenderness, or disfigurement (plexiform type); neurological deficits if deep or spinal involvement [2]	Mixed cellularity (Schwann cells, fibroblasts, perineural cells, mast cells) in myxoid stroma; wavy nuclei, “shredded-carrot” collagen [32]	S100 positivity in ~50% of cells; CD34-positive spindle fibroblasts (“fingerprint” pattern); occasional EMA in perineural cells; neurofilament in intratumoral axons [32]
**Schwannoma**	Solitary or multiple, can be asymptomatic or cause pain/numbness/tingling; vestibular schwannoma → hearing loss, tinnitus, vertigo [20,30]	Encapsulated proliferation of Schwann cells with Antoni A (Verocay bodies) and Antoni B areas; absence of intralesional axons [20,32]	Diffuse, strong S100 positivity; calretinin+; CD56+; CD34 negative [32,33]
**MPNST**	Rapidly enlarging painful mass, often in NF1 patients; new neurological deficits; systemic symptoms rare [21,31]	Hypercellular spindle cells in herringbone/fasciculated patterns, pleomorphism, high mitotic rate, necrosis [21,32]	Variable S100 (50–70%, often decreased in high-grade); p53+ in ~75%; EGFR+ in ~35%; loss of H3K27me3 [21,32]
**Lipoma**	Soft, doughy, mobile subcutaneous mass; usually asymptomatic unless compressive [34,35]	Mature adipocytes with thin fibrous septa; often encapsulated [35]	Typically vimentin+; no diagnostic S100 or CD34 staining pattern described
**Epidermal Inclusion Cyst**	Firm, freely movable nodules with central punctum; may become inflamed/infected, painful, discharge keratinous material [23]	Cyst lined by stratified squamous epithelium with granular layer; lumen filled with laminated keratin [23]	Not routinely characterized by IHC [23]
**DFSP**	Indurated, slowly growing dermal plaque or nodule, often on trunk or proximal limbs; may mimic a bruise [24]	Uniform spindle cells in storiform (cartwheel) pattern; honeycomb infiltration of subcutis [24]	Strong CD34+; vimentin+; S100–, factor XIIIa– [24]
**Traumatic Neuroma**	Painful/tender firm nodule at site of prior nerve injury or surgery [25]	Disorganized bundles of nerve fascicles (axons), Schwann cells and fibroblasts within collagenous scar [25]	S100+ in Schwann cells [25]
**Granular Cell Tumor**	Solitary, painless, slow-growing nodules (commonly tongue, head/neck); may be multiple in syndromes [36,37]	Large polygonal cells with small nuclei, abundant eosinophilic granular cytoplasm; Pustulo-ovoid bodies of Milian; pseudoepitheliomatous hyperplasia of overlying epidermis [37]	Strong S100+; vimentin+; CD68 variably+; PAS-positive granules (diastase resistant) [37]
**Desmoid Tumor**	Firm, sometimes painful mass in abdomen or extremities; associated with FAP or prior surgery/pregnancy [26]	Bland fibroblasts/myofibroblasts in sweeping fascicles; infiltrative borders; low mitoses [26]	Nuclear β-catenin+; vimentin+; SMA variably+; S100– [26]
**Dermal Neurotized Melanocytic Nevus**	Pigmented dermal papule or nodule; may mimic neurofibroma clinically [27]	Neurotized nevus cells in dermis; may have increased mast cells [27]	Melan-A/MART-1 strong+ in nevus cells; S100+; neurofibromas are Melan-A [27]
**Aggressive Angiomyxoma**	Deep perineal/pelvic mass in women of child-bearing age; often asymptomatic viscerally [28,38,39]	Low cellularity spindle cells in myxoid stroma with numerous blood vessels; infiltrative edges [28]	Estrogen and progesterone receptor+; vimentin+; desmin variably+ [28]

## Data Availability

The original contributions presented in this study are included in the article. Further inquiries can be directed to the corresponding author.

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
