# Peer review of "Giant Atypical Neurofibroma of the Calf in Neurofibromatosis Type 1: Case Report and Literature Review"

_reports, 2025, doi:10.3390/reports8030112_

Round 1
Reviewer 1 Report
Comments and Suggestions for Authors
In this manuscript, Lyubomir et al. reported reports a rare case of a 33-year-old male with Neurofibromatosis Type 1 who developed a giant and painful ANNUBP. They then follow-up at 30 months postoperatively confirmed no local recurrence, validated the surgical efficacy for local control. The case report is suitable to publish in our journal Reports. However, before considering to publish in our journal, some revisions must be made.
- As mentioned in Line 46, “ANNUBP lesions exhibit atypical histological features and often harbour CDKN2A/B loss”, The authors need to detect CDKN2A/B loss of this case. CDKN2A/B loss is the key factor of ANNUBP in WHO 2020
- Please provide the information of Surgical Margin Status (Negative/Positive)
- In line 141-151, the authors have described the “giant neurofibroma”, please use the relative size standards (such as tumor volume/anatomical chamber volume ratio) instead of subjective descriptions.
- Please add measuring scale in Figure 2 and Figure 3, and add the enlarged illustrations of key areas, such as atypical cell regions
- Please use ANNUBP throughout the manuscript, to avoid mixing “atypical neurofibroma”.
Author Response
Dear Editors and Reviewers,
Thank you for your thorough review and the insightful feedback on our manuscript, "Giant Atypical Neurofibroma of the Calf in Neurofibromatosis Type 1: Case Report and Literature Review" (ID: reports-3684902). We appreciate the time you have dedicated to improving our work. We have carefully addressed all comments and have revised the manuscript accordingly. As noted, Figures 1 and 2 have been updated based on your suggestions.
Please find our point-by-point responses below.
Response to Reviewer 1
- Comment: As mentioned in Line 46, “ANNUBP lesions exhibit atypical histological features and often harbour CDKN2A/B loss”, The authors need to detect CDKN2A/B loss of this case. CDKN2A/B loss is the key factor of ANNUBP in WHO 2020.
Response: We thank the reviewer for highlighting this critical point. We fully agree that homozygous loss of CDKN2A/B is a key molecular hallmark of ANNUBP. Unfortunately, this specific genetic analysis was not available at our institution at the time of the patient's diagnosis. The diagnosis was therefore established based on the constellation of histopathological and immunohistochemical features, including hypercellularity, moderate cytologic atypia, and loss of the typical neurofibroma architecture, which meet the consensus criteria for ANNUBP. We have added a sentence in the Discussion to acknowledge this limitation and note the importance of molecular testing in modern diagnostics. - Comment: Please provide the information of Surgical Margin Status (Negative/Positive). Response: Thank you. This information is included in the "Case Presentation" section. We state: "Wide resection with negative margins was performed...". We have ensured this is clearly presented in the revised manuscript.
- Comment: In line 141-151, the authors have described the “giant neurofibroma”, please use the relative size standards (such as tumor volume/anatomical chamber volume ratio) instead of subjective descriptions.
Response: We appreciate this suggestion. We have included a detailed paragraph in the "Discussion" section addressing the definition of a "giant" tumor. We argue that using relative size standards, such as a ratio of tumor volume to body mass, can be problematic and potentially misleading (e.g., a tumor of the same absolute size would be considered "less giant" in a larger patient). We have therefore proposed what we believe is a more clinically relevant definition: "the term 'giant' should be applied to lesions that either approach or exceed the size of their anatomical area...". We believe this standard better reflects the surgical challenges and is less prone to ambiguity. - Comment: Please add measuring scale in Figure 2 and Figure 3, and add the enlarged illustrations of key areas, such as atypical cell regions.
Response: We have acted on this valuable suggestion. Scale bars have now been added to all panels in the revised Figures 2 and 3. Furthermore, we have provided several high-magnification views (e.g., Figure 2, panels C and D; Figure 3, all panels) to clearly illustrate the key histological features and atypical cellular regions. - Comment: Please use ANNUBP throughout the manuscript, to avoid mixing “atypical neurofibroma”.
Response: We agree completely. For consistency, we have revised the manuscript to use the term "Atypical Neurofibromatous Neoplasm of Uncertain Biologic Potential (ANNUBP)" throughout. The synonym "atypical neurofibroma (AN)" is now only mentioned at its first use, in line with the latest WHO classification.
Reviewer 2 Report
Comments and Suggestions for Authors
The manuscript describes an interesting clinical case of malignant neurofibroma. The authors provided a detailed description of the diagnostic search, surgical and therapeutic measures. The discussion describes in detail the arguments on the effectiveness and problems of diagnosis and classification of neurofibromatosis. The work may be of interest to the reader, however, I think it is necessary to supplement the manuscript with the following fragments:
• In Figure 3, you need to add a scale scale.
• * If possible, it is advisable to present histological photographs in greater magnification and detail.
• I also recommend attaching digital scans of histopathological preparations to the manuscript as an Appendix.
• I would recommend supplementing the manuscript in the "Discussion" section with information about similar pathological conditions that neurofibroma can disguise itself as.
• It is advisable for the authors to indicate why the ontogenetic profiling of the tumor was not performed.
Author Response
Response to Reviewer 2
- Comment: In Figure 3, you need to add a scale scale.
Response: Thank you. A scale bar has been added to each panel in the revised Figure 3. - Comment: If possible, it is advisable to present histological photographs in greater magnification and detail.
Response: We have ensured that the images in Figure 3 are at a high magnification (20 µm scale bar) to provide significant cellular detail. - Comment: I also recommend attaching digital scans of histopathological preparations to the manuscript as an Appendix.
Response: Thank you for your thoughtful suggestion. While we appreciate the value of including digital histopathology scans, we’ve instead revised our figures 2 and 3 to include the most representative histopathological demonstration. Therefore, we believe adding digital scans would be redundant. - Comment: I would recommend supplementing the manuscript in the "Discussion" section with information about similar pathological conditions that neurofibroma can disguise itself as.
Response: We thank the reviewer for this important point. We have now included a new paragraph in the "Discussion" section dedicated to the differential diagnosis of ANNUBP, addressing its distinction from entities like schwannoma and low-grade MPNST. To further clarify this for the reader, we have also created a new comparative table (Table 1) that summarizes the key distinguishing features. - Comment: It is advisable for the authors to indicate why the ontogenetic profiling of the tumor was not performed.
Response: Thank you for this question. Unfortunately, comprehensive ontogenetic profiling, including molecular analysis for CDKN2A/B loss, was not performed due to limitations in its availability at our institution at the time. We have added a statement to the discussion to clarify this.

Reviewer 3 Report
Comments and Suggestions for Authors
In this case report, Gaydarski et al. describe a 33-year-old male with neurofibromatosis type 1 (NF1) who developed a rapidly enlarging, painful mass in his right calf, ultimately diagnosed as an atypical neurofibroma/ANNUBP based on imaging and pathological features and successfully treated with wide surgical resection, resulting in no recurrence at 30 months and highlighting key diagnostic and management strategies for these premalignant lesions. The following points need to be addressed:
- Please include scale bar in Figures 2 and 3.
- In line 87, how has the proliferation index been calculated?
- In line 85 to 87, can the authors provide information about S100, p53 and CD34 markers and their relevance to this study?
- There is extra irrelevant text in the caption of Figure 3 which needs to be removed.
- Can the authors summarize the ANNUBP cases mentioned in the discussion and from the literature (size, location, outcome) to situate this case within the broader clinical landscape?
- In line 147, can the authors describe what they mean by larger patients and thinner patients?
- Can the authors comment on how this case might inform future guidelines or thresholds for surgical intervention in ANNUBP?
Author Response
Response to Reviewer 3
- Comment: Please include scale bar in Figures 2 and 3.
Response: We have added scale bars to all panels in the revised Figures 2 and 3 and have updated the captions accordingly. - Comment: In line 87, how has the proliferation index been calculated?
Response: Thank you for the query. The method is described in the revised "Case Presentation" section. The Ki-67 proliferation index was "calculated as the ratio of Ki-67 positive cells to all cells per 2mm²". - Comment: In line 85 to 87, can the authors provide information about S100, p53 and CD34 markers and their relevance to this study?
Response: We appreciate the opportunity to clarify the relevance of these markers. The revised "Case Presentation" and "Discussion" sections detail their significance:
- S100: Focal positivity confirms the Schwannian lineage, but its scarcity is a key feature indicating a loss of typical neurofibroma architecture, which supports the ANNUBP diagnosis.
- p53: The negative expression is relevant as it suggests a less aggressive biology compared to high-grade MPNSTs, which often show strong, diffuse positivity.
- CD34: Loss of the typical CD34-positive fibroblastic network is a criterion for ANNUBP, and the focal positivity observed in our case is consistent with this architectural change.
- Comment: There is extra irrelevant text in the caption of Figure 3 which needs to be removed.
Response: Thank you for your careful reading. We have reviewed and edited the caption for Figure 3 to ensure it is concise and directly relevant to the images presented. - Comment: Can the authors summarize the ANNUBP cases mentioned in the discussion and from the literature (size, location, outcome) to situate this case within the broader clinical landscape?
Response: We thank the reviewer for this thoughtful suggestion. However, a detailed tabulation of all reported ANNUBP cases (including size, location, and outcome) lies beyond the intended scope and length constraints of our current manuscript, which focuses on the singular clinical, radiologic, and histopathologic nuances of our presented case. To avoid diluting our case report, we have opted to discuss representative series and illustrative examples in the text with precise citations [9,10,15]. In our opinion the suggested table would better fit a systematic review on the topic. - Comment: In line 147, can the authors describe what they mean by larger patients and thinner patients?
Response: Thank you for pointing out this ambiguity. We were referring to patients with a higher versus lower body mass index (BMI). We have edited the sentence to make this explicit. - Comment: Can the authors comment on how this case might inform future guidelines or thresholds for surgical intervention in ANNUBP?
Response: We appreciate the reviewer’s interest in the broader implications of our findings. However, as a single‐patient case report of a giant ANNUBP, our primary goal is to document the clinical presentation, diagnostic workup, and surgical management of this rare entity. Developing or revising formal surgical guidelines or threshold criteria would require analysis of a large series of ANNUBP cases with long‑term follow‑up—data that go beyond the scope of our report. To address the reviewer’s underlying concern, we have emphasized in the discussion that the unusually large size (20 cm) and atypical features in our patient highlight the need for individualized decision‑making, multidisciplinary consultation, and careful weighing of functional risk versus oncologic safety. We suggest that future studies aggregating similar cases could indeed inform consensus recommendations, but doing so lies outside the remit of this singular case description. We trust that readers will interpret our findings as an illustrative example rather than prescriptive guidance.
We thank the reviewers once again for their constructive and helpful comments. We are confident that the manuscript has been significantly improved by these revisions and hope it is now suitable for publication in Reports.
Round 2
Reviewer 1 Report
Comments and Suggestions for Authors
The qualify of manuscript have been improved a lot, now it can be accept for publication.
Reviewer 2 Report
Comments and Suggestions for Authors
All comments have been eliminated
Reviewer 3 Report
Comments and Suggestions for Authors
The authors addressed the comments raised earlier.